# Exploring One-Shot Federated Learning by Model Inversion and Token Relabel with Vision Transformers

## Abstract

One-Shot Federated Learning, where a central server learns a global model over a network of federated devices in a single round of communication, has recently emerged as a promising approach. For extremely Non-IID data, training models separately on each client results in poor performance, with low-quality generated data that are poorly matched with ground-truth labels. To overcome these issues, we propose a novel Federated Model Inversion and Token Relabel (FedMITR) framework, which trains the global model by better utilizing all patches of the synthetic images. FedMITR employs model inversion during the data generation process, selectively inverting semantic foregrounds while gradually halting the inversion process of uninformative backgrounds. Due to the presence of semantically meaningless tokens that do not positively contribute to ViT predictions, some of the generated pseudo-labels can be utilized to train the global model using patches with high information density, while patches with low information density can be relabeled using ensemble models. Extensive experimental results demonstrate that FedMITR can substantially outperform existing baselines under various settings.

## 1 Introduction

Federated learning (FL) (McMahan et al., 2017) is a machine learning framework where multiple clients collaborate to solve machine learning problems under the coordination of a central server or service provider. And the raw data of each client is stored locally and not exchanged or transferred (Konecnỳ et al., 2016). Recent years, FL has shown its potential to facilitate real-world applications in many fields, including recommender systems (Liang et al., 2021; Liu et al., 2021b), medical image analysis (Liu et al., 2021a; Chen et al., 2021), computer vision (Lu et al., 2023; Zhang et al., 2023), and natural language processing (Zhu et al., 2020; Deng et al., 2022). However, FL poses significant challenges in terms of communication cost and data heterogeneity across clients. Communication cost is a major bottleneck in FL systems, as clients need to communicate frequently with the server over multiple rounds during the training process. This paradigm brings forth significant challenges: 1) heavy communication burden (Li et al., 2020), 2) the risk of connection drop errors between clients and the server (Kairouz et al., 2021; Dai et al., 2022), and 3) potential risk for man-in-the-middle attacks (Wang et al., 2021) and various other privacy or security concerns (Mothukuri et al., 2021; Yin et al., 2021).

One-shot FL (Guha et al., 2019) has emerged as a solution to these issues by restricting communication rounds to a single iteration, thereby mitigating errors arising from multi-round communication and concurrently diminishing the vulnerability to malicious interception. Furthermore, one-shot FL framework is particularly within contemporary model market scenarios (Vartak et al., 2016) where clients predominantly offer pre-trained models. The drawbacks of this method stem from its challenges when dealing with strongly non-iid data (Beitollahi et al., 2024). Due to its single communication session, it is unable to indirectly gather information from other clients through multiple interactions. This leads to a significantly low and unstable accuracy in the single aggregation, as each client's acquired knowledge is extremely limited.

Existing methods often address this challenge by employing federated distillation to acquire knowledge. The server model is aggregated by distilling knowledge from all client models, commonly using

the ensemble, while the ensemble is also responsible for synthesizing data samples for Data-Free Knowledge Distillation (DFKD) (Zhang et al., 2022a;b). We conducted an in-depth analysis and rethink of existing methods, and found that in the setting of highly heterogeneous data in FL, the knowledge obtained from training various local client models is extremely disparate. As a result, the quality of data generated by simple generators is poor, and there are many cases where labels do not match the data.

To address these challenges, we propose a novel one-shot FL framework named FedMITR which trains the global model by better utilizing all patches of the generated images. FedMITR is based on the model inversion framework for data synthesis on the server side, utilizing the ViT model. We start by synthesizing input images from random noise, without utilizing any additional information from the training data, making it suitable for FL where data privacy is crucial. After a single communication round, we obtain only the model without any data. In a data-free scenario, our approach involves recovering training data from pre-trained client models in some manner and utilizing it for knowledge transfer. Furthermore, due to the dispersed training of client models, the knowledge from each client is limited, resulting in poor quality of synthesized data. Therefore we need to select sparse patches with different information densities for subsequent processing. For patches with high information density, they are likely to match pseudo-labels and can be directly used for training the global model. For patches with low information density, we also reuse them by relabeling through ensemble models for knowledge distillation. The experimental results indicate that FedMITR significantly improves accuracy compared to existing one-shot FL methods across various heterogeneous data scenarios. For example, FedMITR surpasses the best baselines with 3.20%, 8.92%, and 7.93% on CIFAR10, OfficeHome, and Mini-Imagenet under $Dir(0.1)$ heterogeneous setting, respectively.

In summary, our main contributions are summarized as follows:

- We rethink the limitations of existing DFKD methods in FL and first explore the role of vision transformers and model inversion in one-shot FL.
- We propose a novel federated model inversion and token relabel framework named FedMITR. In the model inversion stage, we invert well-trained local models to synthesize images with sparse tokens starting from random noise. And in the token relabel stage , we also utilize the role of other tokens encoding some information on all generated image patches.
- Our proposed method FedMITR is only improved for the server side, requiring no additional training on local clients, making it suitable for contemporary model market scenarios without the need for extra data or model transmissions.
- Extensive analytical and empirical studies on various datasets verify the effectiveness of our proposed FedMITR, consistently outperforming other baselines.

## 2 RELATED WORK

**One-Shot Federated Learning.** Guha et al. (Guha et al., 2019) first propose the concept of One-Shot Federated Learning, which treats local models as an ensemble for final prediction and further introduced the use of knowledge distillation along with public data for this ensemble in a single round of communication. Zhou et al. (Zhou et al., 2020) refrain from using public data and instead propose transmitting refined local datasets to the server. Li et al. (Li et al., 2021) propose a method utilizing a two-tier knowledge transfer structure FedKT for distillation on public datasets. Instead of using public data, Zhang et al. (Zhang et al., 2022a) propose a data-free method for knowledge distillation by synthesizing data directly from ensemble models on the server-side. Diao et al. (Diao et al., 2023) and Heinbaugh et al. (Heinbaugh et al., 2023) modify the local training phase and by introducing placeholders or conditional variational autoencoders require additional transmissions. Yang et al. (Yang et al., 2024) suggest using auxiliary pre-trained diffusion models. Previous works either required additional transmission of information from clients or trained simple generators to synthesize low-quality data, which cannot cope with one-shot FL settings in extremely heterogeneous environments.

**Model Inversion.** Fredrikson et al. (Fredrikson et al., 2015) introduce model inversion attack to reconstruct private inputs. Subsequent works broaden this approach to new attack scenarios (He et al., 2019; Yang et al., 2019) . Morerecently, model inversion has been used in data-inaccessible scenarios for tasks like data-free knowledge transfer (Yu et al., 2023; Braun et al., 2024; Patel et al., 2023).

DeepInversion (Yin et al., 2020b) improves synthetic data with batch norm distribution regularization for visual interpretability. However, previous studies don't utilize model inversion in FL and it's merely used as a tool for synthesizing surrogate data. Our work is the first to apply it to one-shot FL and enhance it to obtain generated data that can be better utilized.

**Vision Transformer.** ViT (Dosovitskiy et al., 2020) is one of the earlier attempts that achieved state-of-the-art performance on ImageNet classification, using pure transformers as basic building blocks (Vaswani et al., 2017). DeiT (Touvron et al., 2021) manages to tackle the data-inefficiency problem by simply adjusting the network architecture and adding an additional token along with the class token for Knowledge Distillation to improve model performance. In this paper, we focus on using existing ViT models to distinguish high and low information density tokens, aiming for better knowledge transfer from ensemble models to the global model.

The most related works to our is the DENSE (Zhang et al., 2022a) and DeepInversion (Yin et al., 2020b). In one-shot FL, we applied a new model inversion method for data synthesis, which diverges from traditional DFKD approaches. Traditional methods, such as DENSE (Zhang et al., 2022a), generate synthetic data for distillation by training generators. In contrast, we do not require training generators but instead directly use local models to invert and synthesize data. Unlike (Yin et al., 2020b), which uses entire images generated by inversion for knowledge transfer, our method distinguishes tokens into high information density tokens and low information density tokens and relabels the latter for better utilization of the synthetic data.

# 3 RETHINKING THE DATA-FREE METHOD IN ONE-SHOT FL

In this section, we first review the basic process of One-Shot Federated Learning. Then, we rethink the shortcomings of existing data-free methods in synthesizing pseudo data for FL.

## 3.1 PRELIMINARY

We focus on the centralized setup that consists of a central server and a set of clients $\mathbb{C}$, with $N = |\mathbb{C}|$ clients owning private labeled datasets $\mathbb{D} = \{(\boldsymbol{X}_i, \boldsymbol{Y}_i)\}_{i=1}^N$ in total, where $\boldsymbol{X}_i = \{(\boldsymbol{x}_i^k)\}_{k=1}^{n_i}$ follows the data distribution $\mathcal{D}_i$ over feature space $\mathcal{X}_i$, i.e., $\boldsymbol{x}_i^k \sim \mathcal{D}_i$ and $\boldsymbol{Y}_i = \{(y_i^k)\}_{k=1}^{n_i}$ denotes the ground-truth labels of $\boldsymbol{X}_i$. The goal of one-shot federated learning is to train a good machine learning model $\boldsymbol{f}_S(\cdot)$ with parameter $\boldsymbol{\theta}_S$ over $\mathbb{D} \triangleq \cup_{i=1}^N \mathbb{D}_i$ in only one communication, as in

$$\min_{\boldsymbol{\theta}_S \in \mathbb{R}^d} \mathcal{L}(\theta_S) \triangleq \frac{1}{|\mathbb{D}|} \sum_{i=1}^N \mathbb{E}_{(\boldsymbol{x}_i, \boldsymbol{y}_i) \sim \mathcal{D}_i}[\ell(f_S(\boldsymbol{x}_i; \boldsymbol{\theta}_S), y_i))], \tag{1}$$

where $\ell(\cdot, \cdot)$ is the loss function, $f_S(\boldsymbol{x}_i; \boldsymbol{\theta}_S)$ is the prediction function of the server that outputs the logits (i.e., outputs of the last fully connected layer) of $\boldsymbol{x}_i$ given parameter $\boldsymbol{\theta}_S$ and $y_i$ denotes the corresponding one-hot label of $\boldsymbol{x}_i$.

In FL, the global model is updated by averaging the model parameters from different clients during training. However, this can only be done directly if all the models have the same structure and size, which can be a restrictive constraint in many cases. Additionally, in real-world scenarios, the data distributions across different clients may be Non-IID (Non-Independent and Identically Distributed) or subject to domain shifts. As a result, the global model obtained by averaging model parameters tends to have poor generalization performance. For one-shot FL, it is crucial to aggregate multiple local models into a single global model. Ensemble learning allows combining multiple heterogeneous weak classifiers by averaging the predictions of individual models. In FL, the original training set $\mathbb{D}_i$ cannot be accessed, and only well-pretrained models $\boldsymbol{f}_i(\cdot)$ parameterized by $\boldsymbol{\theta}_i$, are provided. Here, we define the Ensemble $\boldsymbol{E}_S(\cdot)$ as:

$$\boldsymbol{E}_S(\boldsymbol{x}; \{\boldsymbol{\theta}_i\}_{i=1}^N) \triangleq \sum_{i=1}^N w_i \boldsymbol{f}_i(\boldsymbol{x}; \boldsymbol{\theta}_i), \tag{2}$$

where $\boldsymbol{f}_i(\boldsymbol{x}; \boldsymbol{\theta}_i)$ is the prediction function that output the logits of $\boldsymbol{x}$ given the model $\boldsymbol{\theta}_i$, while $\boldsymbol{w} = [w_1, w_2, .., w_N]$ adjusts the weights of each local client logits. Typically we set $w_i = 1/N$, especially for the server that do not know the number of data points for each client. And We use $\boldsymbol{E}_S(\boldsymbol{x})$ to denote $\boldsymbol{E}_S(\boldsymbol{x}; \{\boldsymbol{\theta}_i\}_{i=1}^N)$, which means the output logits of the Ensemble given $\boldsymbol{x}$.

## 3.2 RETHINKING THE WAY OF SYNTHESIZING DATA IN FL

**How do traditional synthetic data methods work?** To tackle the problems in one-shot FL as mentioned in Section 3.1, many previous works (Zhang et al., 2022a;b; Dai et al., 2024) utilize server-side knowledge distillation to improve the global model without the need to share additional information or rely on any auxiliary dataset. They primarily focus on designing loss functions and training a generator on the server side to generate data, which is then used for knowledge transfer with ensemble models. More detailed information is provided in the Appendix.

**What are the challenges of traditional synthetic data methods?** Although the performance of the server-side model has been improved through traditional methods, as seen from Figure 1(b), the synthesized data distribution shows no clear boundaries. The use of such low-quality data for federated distillation limits the scope for performance enhancement. This is because, on one hand, previous efforts mainly involved training a generator to synthesize data using relatively simple generator model structure. On the other hand, in the face of the highly heterogeneous challenges in federated learning data, many generated data labels do not match the data. This results in a large number of errors being learned by the global model during subsequent distillation, thereby limiting performance improvement.

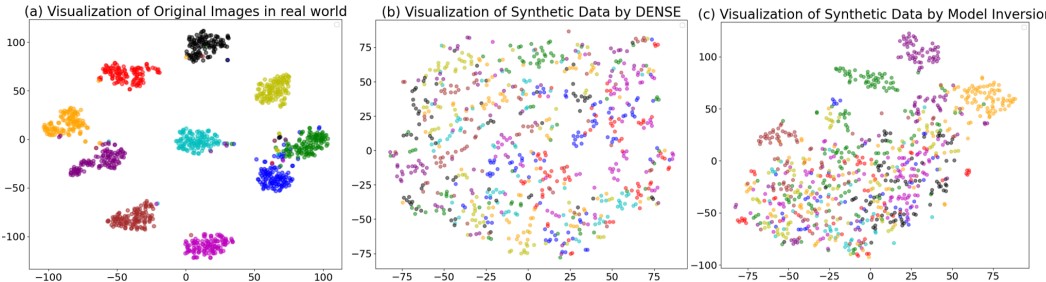

Figure 1: t-SNE visualisation of the features. (a)-(c) represent the feature distribution visualizations for the original training images, the synthetic images using traditional methods, and the synthetic images using model inversion methods on CIFAR10, respectively.

Inspired by the findings of the drawbacks of traditional methods mentioned above, we first consider incorporating ViTs and model inversion methods into one-shot FL to enhance the quality of generated data. As shown in Figure 1(c), samples generated through model inversion methods exhibit more distinct boundaries in data distribution compared to traditional methods. Furthermore, due to the potential mismatch between pseudo-labels and data content, we also further process the tokens of generated data. During generation, we selectively filter out patches with higher weights, and in the subsequent distillation phase, we separately relabel patches with lower weights.

## 4 METHODOLOGY

To overcome the shortcomings of traditional synthetic data methods in one-shot FL mentioned in Section 3.2, we propose a novel federated framework named FedMITR and the illustration of the training process in the server is demonstrated in Figure 2. After clients upload their well-trained local models to the server, we first invert well-trained networks (local models) to synthesize class-conditional images starting from random noise without using any additional information on the training dataset due to privacy concernsn in the model inversion stage. Next, we use patches with high information density, accompanied by generated pseudo-labels, to assist in training the global model, while relabeling patches with low information density for knowledge distillation. These two stages iterate multiple times on the server side.

### 4.1 MODEL INVERSION

Our goal is to invert well-trained local models to synthesize images starting from random noise without using any additional training data. Additionally, our goal is to not leak any private information

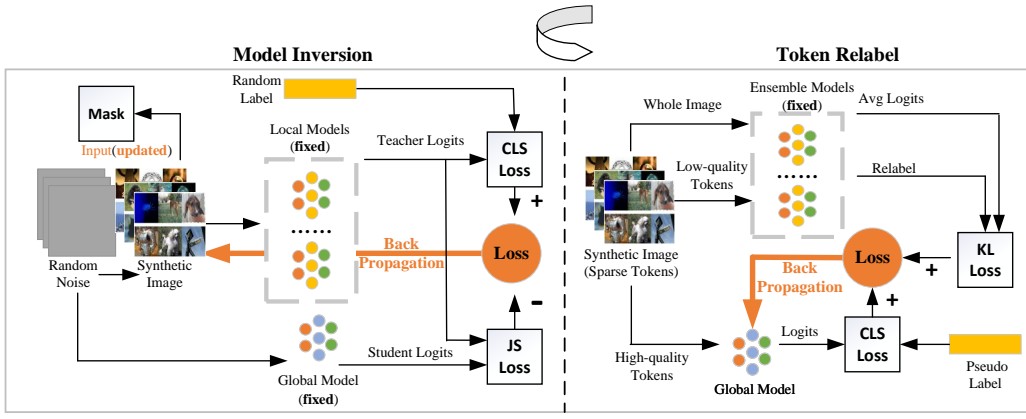

Figure 2: An illustration of server training process of FedMITR, which consists of two stages: (1) In the model inversion stage, we invert well-trained local models to synthesize input images with sparse tokens starting from random noise. (2) In the token relabel stage , we fully utilize the role of other patch tokens encoding some information on all generated image patches.

from the data we generate, meaning attackers cannot predict any sensitive information of clients from the generated data. When traditional inversion methods are applied to ViT, all patches will undergo inversion. Therefore, we refer to this process as dense model inversion with redundant computation and unintended inversion of spurious correlations. In some models, many pieces of information in the inverted images are redundant or incorrect. In contrast, we adopt sparse model inversion, where only image patches with high-density information are inverted, while those without semantic information are filtered out through masking.

To ensure the diversity of generated data, we utilize all pretrained models from the clients for model inversion. Given the local model $\boldsymbol{f}_i(\cdot)$ parameterized by $\boldsymbol{\theta}_i$ and the server model $\boldsymbol{f}_S(\cdot)$ parameterized by $\boldsymbol{\theta}_S$, a randomly initialized input with a new feature distribution $\hat{\boldsymbol{x}} \in \mathbb{R}^{H \times W \times C}$ (height, width, and number of channels) and a random uniformly sampled label $\hat{y}$. The model inversion process involves optimizing a classification loss, a Jensen-Shannon (JS) divergence loss with a negative scaling factor $\alpha$, and a regularization term:

$$\min_{\hat{\boldsymbol{x}}} \mathcal{L}_{\mathrm{MI}} = \mathcal{L}_{\mathrm{CLS}}\left(\boldsymbol{\theta}_i(\hat{\boldsymbol{x}}), \hat{y}\right) + \alpha \mathcal{L}_{\mathrm{JS}}\left(\boldsymbol{\theta}_i(\hat{\boldsymbol{x}}), \boldsymbol{\theta}_S(\hat{\boldsymbol{x}})\right) + \mathcal{R}(\hat{\boldsymbol{x}}), \tag{3}$$

where $\mathcal{L}_{\mathrm{CLS}}(\cdot)$ is a classification loss (e.g., cross-entropy loss) to ensure the label-conditional inversion, which desires $\hat{\boldsymbol{x}}$ could be predicted as $\hat{y}$ and exhibit discriminative features of $\hat{y}$. $\mathcal{R}(\cdot)$ is an prior image regularization term to steer $\boldsymbol{x}$ away from unrealistic images with no discernible visual information, used to penalize the total variance for local consistency (Dosovitskiy & Brox, 2016):

$$\mathcal{R}_{\mathrm{prior}}(\hat{\boldsymbol{x}}) = \alpha_{\mathrm{tv}} \mathcal{R}_{\mathrm{tv}}(\hat{\boldsymbol{x}}) + \alpha_{\ell_2} \mathcal{R}_{\ell_2}(\hat{\boldsymbol{x}}), \tag{4}$$

where $\mathcal{R}_{\mathrm{tv}}$ and $\mathcal{R}_{\ell_2}$ are the total variance and $\ell_2$ norm, respectively, with scaling factors $\alpha_{\mathrm{tv}}, \alpha_{\ell_2}$.

In Figure 2, the mask method refers to dividing the synthesized data generated by model inversion into two parts: tokens with high information density and tokens with low information density. The first question to address is how to identify the semantic patches crucial for inversion. In ViT, the input image $\boldsymbol{X}$ is projected to three matrices, namely query $\boldsymbol{Q}$, key $\boldsymbol{K}$, and value $\boldsymbol{V}$ matrices. The attention operation is defined as (Vaswani et al., 2017):

$$\mathrm{Attention}(\boldsymbol{Q}, \boldsymbol{K}, \boldsymbol{V}) = \mathrm{Softmax}\left(\frac{\boldsymbol{Q}\boldsymbol{k^T}}{\sqrt{d}}\right)\boldsymbol{V}, \tag{5}$$

where $d$ is the length of the query vectors in $\boldsymbol{Q}$. We define the softmax output matrix in Eq.(5) as the square matrix $\boldsymbol{A}$, which is known as the attention map, representing attention weights of all token pairs. We define $\boldsymbol{a}_i \triangleq \boldsymbol{A}_{[i,:]}$, $\boldsymbol{a}_i$ indicating the attention weights from $\hat{\boldsymbol{x}}_i$ to all tokens $[\hat{\boldsymbol{x}}_{\mathrm{cls}}, \hat{\boldsymbol{x}}_1, \ldots, \hat{\boldsymbol{x}}_L]$. And at iteration $t$ within the inversion process, we propose to identify semantic patches utilizing the attention weights $\boldsymbol{a}_{\mathrm{cls}}$ from the preceding iteration $t-1$. The output $\hat{\boldsymbol{x}}_{\mathrm{cls}}$ is a

---

**Algorithm 1** Server training process of FedMITR

---

1: **Input:** Clients' local models $\{\boldsymbol{f_1}(), \cdots, \boldsymbol{f_N}()\}$, server model $\boldsymbol{f}_S()$ with parameter $\boldsymbol{\theta}_S$, synthetic dataset $\mathbb{D}_S = \emptyset$, ensemble model $\boldsymbol{E}_S$, learning rate of model inversion and token relabel $\eta_G$ and $\eta_S$, inversion iterations $T_I$, global model training epochs $T$, and batch size $b$

2: **Output:** Server model $\boldsymbol{f}_S()$ with parameter $\boldsymbol{\theta}_S$

3: **for** epoch = 0 to $T - 1$ **do**

4:     *// Model Inversion*

5:     **for** $i = 0$ to $N - 1$ **do**

6:         Sample a batch of noises and labels $\{\boldsymbol{z}_i, y_i\}_{i=1}^b$

7:         **for** $t_i = 0$ to $T_I - 1$ **do**

8:             Generate $\{\hat{\boldsymbol{x}}_i\}_{i=1}^b$ with $\{\boldsymbol{z}_i\}_{i=1}^b$

9:             Update the inputs: $\hat{\boldsymbol{x}} \leftarrow \hat{\boldsymbol{x}} - \eta_G \bigtriangledown_{\hat{\boldsymbol{x}}} \mathcal{L}_{\mathrm{MI}}(\hat{\boldsymbol{x}})$, where $\mathcal{L}_{\mathrm{MI}}(\hat{\boldsymbol{x}})$ is defined in Eq.(3)

10:            Mask $\hat{\boldsymbol{x}}$ by the matrix $\boldsymbol{A}$ is defined in Section 4.1

11:         **end for**

12:     **end for**

13:     $\mathbb{D}_S \leftarrow \mathbb{D}_S \cup \{\hat{\boldsymbol{x}}_i\}_{i=1}^b$

14:     *// Token Relabel*

15:     **for** sampling batch $\{\hat{\boldsymbol{x}}\}$ in $\mathbb{D}_S$ **do**

16:         Update the server model: $\boldsymbol{\theta}_S \leftarrow \boldsymbol{\theta}_S - \eta_S \bigtriangledown_{\boldsymbol{\theta}_S} \mathcal{L}_{\mathrm{TR}}(\boldsymbol{\theta}_S)$, where $\mathcal{L}_{\mathrm{TR}}(\boldsymbol{\theta}_S)$ is defined in Eq.(6)

17:     **end for**

18: **end for**

---

linear combination of all tokens' value vectors, weighted by $\boldsymbol{a}_{\mathrm{cls}}$. Since $\hat{\boldsymbol{x}}_{\mathrm{cls}}$ in the final layer serves for classification, it is rational to view $\boldsymbol{a}_{\mathrm{cls}}$ as an indicator, measuring the extent to which each token contributes label-relevant information to final predictions. We first assess the importance of each remaining token based on the attention weights from the previous iteration $t - 1$. Then, we stop the inversion of the mask ratio $r$ of patches with the lowest attention.

## 4.2 TOKEN RELABEL

Due to the heterogeneity of data in federated learning, many clients' data may not even contain certain classes in extreme cases. Therefore, many synthesized data labels mismatch with the data itself, requiring the relabeling of certain tokens. In such scenarios, the knowledge learned by each client's model is extremely limited. First, we begin by utilizing the overall image, following the approach outlined in (Lin et al., 2020). However, relying solely on distillation loss is insufficient for achieving good results. Not all tokens output by ViTs can be directly and simply utilized (Jiang et al., 2021) and examples include that tokens containing semantically meaningless or distractive image backgrounds do not positively contribute to the ViT predictions (Liang et al., 2022). Therefore, we utilize tokens of varying information densities generated during the model inversion stage. We train the global model with pseudo-labels for tokens with high information density and conduct distillation for tokens with low information density using an ensemble model for relabeling. The overall loss function for the entire second stage is as follows:

$$\min_{\boldsymbol{\theta}_S} \mathcal{L}_{\mathrm{TR}} = \mathcal{L}_{\mathrm{KD}} + \lambda_1 \mathcal{L}_{\mathrm{CLS}} \left( \boldsymbol{\theta}_S(\hat{\boldsymbol{x}}_h), \hat{y} \right) + \lambda_2 \mathcal{L}_{\mathrm{KL}}(\boldsymbol{E}_S(\hat{\boldsymbol{x}}_l), \boldsymbol{f}_S(\hat{\boldsymbol{x}}_l; \boldsymbol{\theta}_S)), \tag{6}$$

where $\mathcal{L}_{\mathrm{KD}}$ is defined in Eq.(8) in the Appendix, $\mathcal{L}_{\mathrm{KL}}(\cdot)$ is a Kullback-Leibler (KL) divergence loss. $\hat{\boldsymbol{x}}_h$ and $\hat{\boldsymbol{x}}_l$ represent the high-density and low-density information tokens respectively, and $\lambda_1, \lambda_2$ are the scaling factors. The two stages are iterated multiple times on the server-side, eventually training a suitable global model for all clients.

## 4.3 OVERALL TRAINING ALGORITHM

First, training is performed on each local client in FL (this paper does not focus on improvements in client-side training methods). Then, all trained local models are transmitted to the server through a single communication round. The server-side training process of FedMITR is shown in Algorithm 1. After finishing the server side training process, we obtain a global model applicable to all clients.

# 5 EXPERIMENTS

In this section, we conduct extensive experiments to verify the effectiveness of our proposed approach. Due to space limitations, part of the experimental setups and results are placed in the **Appendix**.

## 5.1 EXPERIMENTAL SETUP

**Datasets and partitions.** Our experiments are conducted on the following four popular real-world datasets: CIFAR10 (Krizhevsky et al., 2009), CIFAR100 (Krizhevsky et al., 2009), OfficeHome (Venkateswara et al., 2017) and Mini-ImageNet (Vinyals et al., 2016). To simulate real-world applications, we adopt two different kinds of partition: 1) $\boldsymbol{p}_k \sim Dir(\alpha)$: for each class, we allocate a $\boldsymbol{p}_k^i$ proportion of the data of class $i$ to client $k$. The parameter $\alpha$ controls the level of statistical imbalance, with a smaller $\alpha$ inducing more skewed label distributions among local clients. 2) $\#C = k$: each client only has data from $k$ classes and we assign $k$ random classes for each client.

**Baselines.** We compare the performance of FedMITR against four existing FL methods: FedAvg(McMahan et al., 2017), FedFTG (Zhang et al., 2022b), DENSE (Zhang et al., 2022a) and Co-Boosting (Dai et al., 2024). FedAvg(McMahan et al., 2017) and FedFTG (Zhang et al., 2022b) are not proposed for the field of one-shot FL, so they are set to communicate only for a single round in the experiments. Furthermore, since FedMITR is a DFKD method, we also use the data-free method DeepInversion (Yin et al., 2020a) in model inversion as a comparison method, applying it to the one-shot FL setting with ViTs.

**Configurations.** We use DeiT/16-Tiny as train models, which are pre-trained on ImageNet-1K (Russakovsky et al., 2015). All models are accessible from timm and are trained with 10 clients. We perform 100 iterations for model inversion using the Adam optimizer with a learning rate $\eta_G = 0.001$. The image regularization term scaling factor $\alpha_{\text{tv}}$ is set as 1e-4 and the mask ratio $r$ is set as 0.3. The scaling factors $\lambda_1$ and $\lambda_2$ are set to 0.5. For the training of the server model, we use the SGD optimizer with a learning rate $\eta_S = 0.001$. The number of total epochs is set to 50. The distillation temperature $T$ is set to 20. Results are reported across 3 random seeds. Other experimental setups please refer to the Appendix.

## 5.2 GENERAL RESULTS AND ANALYSIS

Table 1: Test accuracy of the server model of different methods on three datasets and across five levels of statistical heterogeneity (lower $\alpha$ is more heterogeneous).

| Dataset | $\alpha$ | FedAvg | FedFTG | DENSE | Co-Boosting | DeepInversion | FedMITR |
|---|---|---|---|---|---|---|---|
| CIFAR10 | 0.01 | 11.70±1.87 | 12.73±2.91 | 12.05±2.30 | 12.07±2.23 | 13.42±1.84 | **19.19±2.33** |
| | 0.05 | 15.87±2.74 | 16.16±2.44 | 16.41±2.36 | 17.31±2.65 | 22.01±2.44 | **26.78±2.12** |
| | 0.1 | 24.30±3.72 | 25.05±4.28 | 25.19±3.12 | 26.88±2.74 | 33.77±2.03 | **36.97±2.98** |
| | 0.3 | 37.93±3.73 | 40.01±5.35 | 39.02±4.05 | 40.57±5.57 | 44.63±3.70 | **51.69±2.86** |
| | 0.5 | 39.37±1.53 | 42.02±2.81 | 40.08±1.79 | 41.55±2.17 | 45.00±2.11 | **49.45±5.86** |
| OfficeHome | 0.01 | 8.06±1.40 | 8.79±1.49 | 8.62±1.40 | 8.82±1.35 | 11.88±1.51 | **24.05±1.19** |
| | 0.05 | 13.68±1.66 | 14.61±1.46 | 14.33±1.42 | 14.70±1.38 | 18.72±0.95 | **30.04±1.47** |
| | 0.1 | 17.63±2.39 | 19.10±1.94 | 18.49±2.19 | 18.82±2.25 | 23.89±1.41 | **32.81±1.55** |
| | 0.3 | 26.88±1.56 | 28.79±1.27 | 28.24±1.47 | 28.60±1.10 | 33.54±1.68 | **35.53±0.32** |
| | 0.5 | 31.13±2.63 | 32.86±2.88 | 32.48±2.77 | 32.91±2.72 | 37.87±3.25 | **38.15±1.63** |
| Mini-ImageNet | 0.01 | 13.99±1.83 | 14.73±2.02 | 14.49±1.84 | 15.08±1.90 | 22.49±1.89 | **45.26±5.14** |
| | 0.05 | 37.98±2.16 | 38.92±2.00 | 38.65±1.95 | 39.03±2.39 | 47.34±2.89 | **62.15±0.28** |
| | 0.1 | 52.48±2.10 | 53.62±1.79 | 53.19±1.87 | 53.47±2.05 | 60.28±2.10 | **68.21±2.01** |
| | 0.3 | 76.84±1.71 | 77.64±1.46 | 77.32±1.71 | 77.46±1.66 | **79.06±1.31** | 77.44±1.50 |
| | 0.5 | 81.87±0.23 | 82.85±0.18 | 82.16±0.08 | 82.21±0.19 | **83.09±0.77** | 82.18±0.22 |

**Overall Comparison.** To evaluate the effectiveness of our method, we conduct experiments under various non-IID settings by varying $\alpha = \{0.01, 0.05, 0.1, 0.3, 0.5\}$ and report the performance across different datasets and methods in Table 1. From the table, we can conclude that FedMITR consistently outperforms all other baselines in all settings, especially in highly heterogeneous scenarios where the Dirichlet distribution parameter is very small. Notably, in many settings, FedMITR achieves over a

significant accuracy improvement compared to the best baseline, DeepInversion. Our approach shows a more significant improvement compared to traditional methods, as these methods do not use ViTs for model training; instead, most methods use CNNs to guide the training of the generator. However, when the heterogeneity is low, such as $\alpha = 0.5$ and $\alpha = 0.3$ on Mini-Imagenet, the accuracy of FedMITR is lower than DeepInversion. This is because when the heterogeneity is low, local models are already well-trained and do not require additional relabeling to facilitate federated distillation. In conclusion, the superiority of our proposed method can be attributed to utilizing the ViT model and model inversion to use all patches, which achieves better utilization of synthesized data.

**Extension to Extreme Heterogeneity.** In Table 2, we use 10 clients, with each client assigned 1 (extreme heterogeneity) and 3 labels in the CIFAR10 dataset with a total of 10 categories, and 10 (extreme heterogeneity) labels in the Mini-ImageNet dataset with a total of 100 categories. In such cases, the accuracy of traditional methods is very low and we can conclude that in settings of extreme heterogeneity, FedMITR can achieve larger improvements compared to traditional methods.

Table 2: Test accuracy of the server model on two datasets under extreme heterogeneity setting.

| Dataset | Partition | FedAvg | FedFTG | DENSE | Co-Boosting | DeepInversion | FedMITR |
|---|---|---|---|---|---|---|---|
| CIFAR10 | #C=1 | 9.10±1.49 | 9.59±2.06 | 9.28±1.60 | 9.37±1.76 | 9.77±1.85 | **13.98±3.46** |
| | #C=3 | 20.12±4.28 | 21.88±1.51 | 20.99±4.22 | 21.26±4.59 | 28.06±4.91 | **33.85±2.48** |
| Mini-ImageNet | #C=10 | 7.80±0.44 | 8.55±0.90 | 8.28±0.66 | 8.39±0.75 | 13.59±0.99 | **21.81±1.22** |

**Effects of the proposed components.** To further assess the effectiveness of FedMITR, which involves model inversion and token relabel, we conduct experiments in different models across four datasets under $Dir(0.1)$ heterogeneous client model setting. And we further study the effectiveness of our proposed components. Table 3 displays four methods: the baseline FedAvg, knowledge distillation based only on model inversion (inversion + KD), training based only on pseudo-labels using model inversion (inversion + PL), and our proposed method FedMITR. The results in the table indicate that, upon obtaining data from model inversion, individually performing knowledge distillation or training with pseudo-labels can enhance the final server model's performance. Moreover, our approach, which combines both strategies and further conducts relabeling followed by distillation on low-information-density tokens, achieves the best performance.

Table 3: Test accuracy of server model in different models across four datasets under $Dir(0.1)$ heterogeneous client model setting.

| Model | Method | CIFAR10 | CIFAR100 | OfficeHome | Mini-ImageNet |
|---|---|---|---|---|---|
| DeiT/16-Tiny | FedAvg | 24.98 | 9.33 | 20.39 | 50.92 |
| | Inversion + KD | 33.25 | 11.08 | 25.44 | 59.19 |
| | Inversion + PL | 34.72 | 11.86 | 34.41 | 66.07 |
| | FedMITR | **35.69** | **13.84** | **34.60** | **68.67** |
| DeiT/16-Base | FedAvg | 57.44 | 30.62 | 39.40 | 80.19 |
| | Inversion + KD | 58.92 | 32.28 | 40.46 | 80.47 |
| | Inversion + PL | 65.93 | 34.61 | 45.98 | **88.52** |
| | FedMITR | **66.54** | **35.19** | **46.70** | 88.37 |
| ViT/16-Small | FedAvg | 50.98 | 12.29 | 37.91 | 78.37 |
| | Inversion + KD | 52.48 | 14.67 | 40.46 | 79.15 |
| | Inversion + PL | 53.50 | 15.87 | 42.24 | 86.18 |
| | FedMITR | **53.82** | **17.66** | **46.07** | **87.88** |

**Different Number of Clients.** We also evaluate the performance of these methods by varying the number of clients participating $N = \{5, 10, 20, 50\}$ in one-shot FL in Table 4. From the table, Although there is a slight decrease in overall performance when increasing the number of clients, FedMITR still achieves the best performance, reaffirming the effectiveness of our approach.

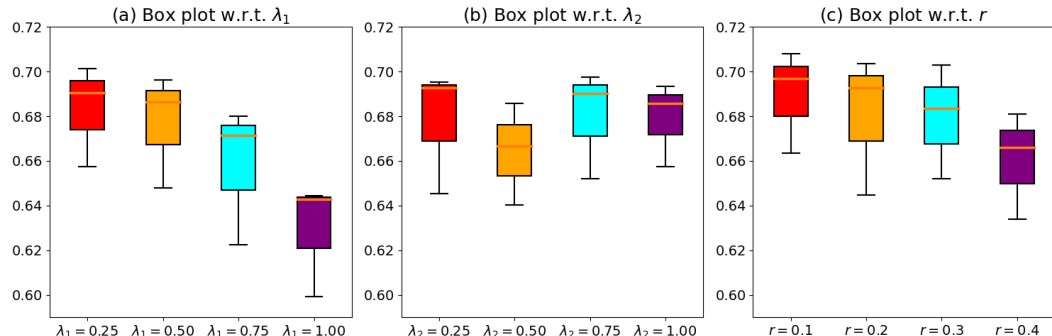

Figure 3: Test accuracy of the server model of FedMITR using different hyper-parameters (a) $\lambda_1$, (b) $\lambda_2$, (c) the mask ratio $r$ on Mini-ImageNet under $Dir(0.1)$ heterogeneous client model setting.

Table 4: Test accuracy of the server model in Mini-ImageNet across different numbers of clients under $Dir(0.1)$ heterogeneous client model setting.

| $N$ | FedAvg | FedFTG | DENSE | Co-Boosting | DeepInversion | FedMITR |
|---|---|---|---|---|---|---|
| 5 | 65.31 | 65.80 | 65.82 | 66.18 | 68.05 | **71.31** |
| 10 | 54.87 | 55.69 | 55.33 | 55.83 | 62.71 | **69.94** |
| 20 | 36.15 | 37.67 | 37.35 | 37.90 | 44.19 | **54.80** |
| 50 | 24.54 | 25.24 | 25.11 | 25.38 | 30.40 | **49.33** |

**Hyperparameters senstivity.** To measure the influence of hyperparameters, we select $\lambda_1$ and $\lambda_2$ from $\{0.25, 0.50, 0.75, 1.00\}$ and select the mask ratio $r$ in $\{0.1, 0.2, 0.3, 0.4\}$. Figure 3 illustrates the test accuracy in term of the box plot, where (a) suggests that an excessively high $\lambda_1$ will lead to a performance drop. This is because too many pseudo-labels are undesirable in heterogeneous conditions as many pseudo-labels do not match the synthetic data. (b) indicates that the parameter $\lambda_2$ is not sensitive, while (c) shows that the mask ratio $r$ should not be too high either.

**Visualization of synthetic data.** To compare the synthetic data (including tokens with high information density and low information density) in our method with the training data, we visualize the synthetic data on the Office-Home and Mini-ImageNet datasets in Figure 4. As shown in the figure, the first/fourth row represents the original data of the OfficeHome/Mini-ImageNet dataset, while the rest are synthetic data generated by models trained on the these datasets. Among them, the second/fifth row consists of selected high-information-density patches, while the third/last row consists of low-information-density patches. Visually, we can not obtain any privacy information because the synthetic images are dissimilar to the original images, effectively reducing the probability of leaking sensitive client information.

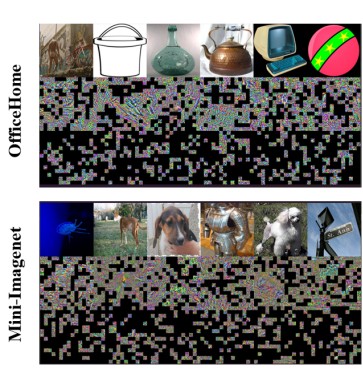

Figure 4: Visualization of synthetic data and training data

## 6 CONCLUSION

In this paper, we first present a comprehensive critique of existing methods using synthetic data in federated learning, emphasizing their drawbacks and limitations. Then, we propose a novel Federated Model Inversion and Token Relabel framework named FedMITR. This framework generates synthetic images through ViTs and efficiently utilizes all tokens of the generated images to train the global model. Extensive analytical and empirical studies on various datasets verify the effectiveness of our method, consistently outperforming other baseline methods under diverse heterogeneous settings.

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

# A APPENDIX

## A.1 MORE DETAILS ABOUT THE DFKD METHODS IN FL

The first step is to train the auxiliary generator during the data generation phase. When aggregating pre-trained models $\{\boldsymbol{\theta}_i\}_{i=1}^N$ into one server model $\boldsymbol{\theta}_S$, we aim to train a generator to generate synthetic data $\mathbb{D}_S$ with the data distribution $\mathcal{D}_S$ based on the Ensemble output. In particular, giving a random noise $\boldsymbol{z}$ generated from a standard Gaussian distribution and a random uniformly sampled one-hot label $\hat{y}$, the generator $G(\cdot)$ with parameter $\theta_G$ is responsible for generating the data $\hat{\boldsymbol{x}} = G(\boldsymbol{z})$, forming the synthetic dataset $\mathbb{D}_S$. Since we are unable to access the training data of clients, we cannot compute the similarity between the synthetic data and the training data directly. Typically, to make sure the synthetic data can be classified correctly with a high probability by the Ensemble $\boldsymbol{E}_S(\cdot)$, as in:

$$\min_{\boldsymbol{\theta}_G \in \mathbb{R}^d} \mathcal{L}(\boldsymbol{\theta}_G) \triangleq \frac{1}{|\mathbb{D}_S|} \sum \mathbb{E}_{\hat{\boldsymbol{x}} \sim \mathcal{D}_S}[\ell_{CE}(\boldsymbol{E}_S(\hat{\boldsymbol{x}}), \hat{y})], \tag{7}$$

where $\ell_{CE}(\cdot, \cdot)$ denotes the cross-entropy function. In addition, various existing data-free methods often incorporate additional loss functions to train generators to ensure the quality of generated data.

After getting the synthetic dataset $\mathbb{D}_S$ based on the well-trained generator in Eq.(7), existing federated distillation methods intends to distill the ensemble $\boldsymbol{E}_S$ into the final server model $\theta_S$ with the help of these synthetic data, as in:

$$\min_{\boldsymbol{\theta}_S \in \mathbb{R}^d} \mathcal{L}(\boldsymbol{\theta}_S) \triangleq \frac{1}{|\mathbb{D}_S|} \sum \mathbb{E}_{\hat{\boldsymbol{x}} \sim \mathcal{D}_S}[\ell_{KL}(\boldsymbol{E}_S(\hat{\boldsymbol{x}}), \boldsymbol{f}_S(\hat{\boldsymbol{x}}; \boldsymbol{\theta}_S))], \tag{8}$$

where $\ell_{KL}(\cdot, \cdot)$ denotes the Kullback-Leibler (KL) divergence.

## A.2 MORE DETAILS ABOUT THE EXPERIMENT

**Dataset.** We use four popular real-world datasets in our experiments. CIFAR10 (Krizhevsky et al., 2009) consists of 60,000 color images in 10 classes, 50,000 for train, and 10,000 for test, while CIFAR100 (Krizhevsky et al., 2009) dataset is similar to the CIFAR10 dataset but it has 100 classes containing 600 images each. OfficeHome (Venkateswara et al., 2017) is a image recognition dataset that includes 15,588 images of 65 classes from four different domains (art, clipart, product, and real-world). Mini-ImageNet (Vinyals et al., 2016) is a small subset extracted from the ImageNet-1K (Russakovsky et al., 2015) dataset, consisting of 100 categories with 600 images per category, totaling 60,000 images. We split 80% data as the training set and 20% of that as the testing set. Each image is resized to a 224×224 color image. All available test data is used to evaluate the final server model.

**Baselines.** To ensure a fair comparison, we disregarded methods that require downloading auxiliary models or additional datasets. Furthermore, due to the single round of communication, regularization-based aggregation methods or similar approaches that rely on multiple iterations are ineffective. Therefore, against four existing FL methods: FedAvg(McMahan et al., 2017), FedFTG (Zhang et al., 2022b), DENSE (Zhang et al., 2022a) and Co-Boosting (Dai et al., 2024). FedAvg(McMahan et al., 2017) learns a shared model by aggregating locally-computed updates and iteratively updating through multiple rounds of communication between clients and the server. FedFTG explores the input space of local models through a generator and uses it to transfer knowledge from the local models to the global model. Additionally, FedFTG proposes a hard sample mining scheme to achieve effective knowledge distillation throughout the training process. Furthermore, FedFTG also develops customized label sampling and class-level ensemble techniques to maximize knowledge utilization, which implicitly mitigates distribution differences among clients. The two methods mentioned above are based on traditional multi-round communication federated learning, so they are set to communicate only once in this paper. DENSE (Zhang et al., 2022a) train a generator that considers similarity, stability, and transferability and performe federated distillation on the server side. Co-Boosting (Dai et al., 2024) uses the current Ensemble to synthesize higher-quality samples in an adversarial manner. These hard samples are then employed to promote the quality of the Ensemble by adjusting the ensembling weights for each client model. Since FedMITR is a DFKD method, we also use the data-free method DeepInversion (Yin et al., 2020a) in model inversion as a comparison method, applying it to the one-shot FL setting. The DeepInversion (Yin et al., 2020a) optimizes the input data while keeping the teacher model fixed during data synthesis, and it regularizes the distribution of intermediate feature

maps using the information stored in the teacher's batch normalization layers. Additionally, adaptive deep inversion is employed to enhance the diversity of the synthesized images, thereby maximizing the Jensen-Shannon divergence between the logits of the teacher and student networks.

**Configurations.** Unless otherwise stated, we conduct experiments with 10 clients. All models are accessible from timm. For each client's training, we use pre-trained DeiT/16-Tiny on ImageNet-1K (Russakovsky et al., 2015) as train models and use the SGD optimizer with learning rate=0.001, momentum=0.9 and weight decay=1e-4. We set the batch size to 64 and the local epoch to 50. We perform 100 iterations for model inversion using the Adam optimizer with a learning rate $\eta_G = 0.001$ and $(\beta_1, \beta_2) = (0.5, 0.99)$ about each local model. The image regularization term scaling factor $\alpha_{tv}$ is set as 1e-4 and the mask ratio $r$ is set to 0.3. The scaling factors $\lambda_1$ and $\lambda_2$ are set to 0.5. The batch size of synthetic data is set to 64. For the training of the global model, we use the SGD optimizer with a learning rate $\eta_S = 0.001$, momentum=0.9 and weight decay=1e-4. The factor $\alpha$ of JS divergence loss is set to 1.0. The framework is implemented with PyTorch and is trained on a single NVIDIA RTX 3090 GPU.

**More details about the ablation experiments.** In our method, the components of the loss function in Eq.(6) collectively form the overall tokens being processed. Therefore, instead of directly removing some components while retaining others for ablation experiments, we employed alternative methods for experimental validation in Table 3. Here, Inversion + KD refers to knowledge distillation based only on model inversion, while Inversion + PL refers to knowledge distillation using only pseudo-labels without employing token relabel. Below is our additional analysis of the hyperparameters. In the face of the highly heterogeneous challenges in Federated Learning, many generated data labels do not match the data. This results in a large number of errors being learned by the global model during subsequent train, thereby limiting performance improvement. So an excessively high $\lambda_1$ will lead to a performance drop. The loss weighted by parameter $\lambda_2$ represents the tokens involved in knowledge distillation through ensemble model re-labeling. Since it is not influenced by potentially erroneous pseudo-labels, it maintains higher robustness and is less sensitive to parameter variations. The Table 5 is the additional ablation experiment about Eq.(6).

Table 5: Ablations on different components of our method in Mini-ImageNet in three random seeds and across three levels of statistical heterogeneity.

| Dataset | $\alpha$ | w/ $\mathcal{L}_{KD}$ | w/o $\mathcal{L}_{CLS}$ | w/o $\mathcal{L}_{KL}$ | FedMITR |
|---------|----------|------|------|------|---------|
| Mini-ImageNet | 0.01 | 22.49±1.89 | 37.34±1.22 | 42.04±1.49 | **45.26±5.14** |
| | 0.05 | 47.34±2.89 | 51.24±1.69 | 57.78±1.00 | **62.15±0.28** |
| | 0.1 | 60.28±2.10 | 61.44±2.47 | 63.12±3.59 | **68.21±2.01** |

**More details about the experiment results.** Due to space limitations and the relatively poor performance of the DeiT-Tiny model on CIFAR-100 (as this paper focuses on server-side improvements and does not use more advanced training methods during local training), we does not include CIFAR-100 results in the main table. Due to overall low performance, we did not conduct in-depth research on other aspects of the experiments. However, our method, FedMITR, is still capable of improving model performance in environments with high heterogeneity on a relative scale in Table 6.

Table 6: Test accuracy of the server model of different methods in CIFAR100 in three random seeds and across three levels of statistical heterogeneity.

| Dataset | $\alpha$ | FedAvg | Co-Boosting | DeepInversion | FedMITR |
|---------|----------|--------|-------------|---------------|---------|
| CIFAR100 | 0.01 | 4.02±0.66 | 4.51±0.57 | 5.89±0.72 | **8.89±1.23** |
| | 0.05 | 6.62±0.78 | 7.33±0.74 | 9.02±0.67 | **11.35±0.89** |
| | 0.1 | 8.55±1.12 | 9.23±1.07 | 10.86±1.60 | **13.12±1.03** |
| | 0.3 | 12.30±0.31 | 13.25±0.75 | 15.24±1.07 | **16.45±1.92** |
| | 0.5 | 13.99±0.82 | 14.87±1.11 | **17.12±1.65** | 17.09±1.90 |

To evaluate the effectiveness of our method, we conduct experiments under various non-IID settings by varying $\alpha = \{0.01, 0.05, 0.1, 0.3, 0.5\}$ and $\#C = k$ in Tables 7 to 11. Below are the detailed results for each random seed.

Table 7: Test accuracy of the server model of different methods in CIFAR10 in three random seeds under extreme heterogeneity setting.

| $\#C = k$ | seed | FedAvg | FedFTG | DENSE | Co-Boosting | DeepInversion | FedMITR |
|---|---|---|---|---|---|---|---|
| | 0 | 7.97 | 7.95 | 7.99 | 7.98 | 8.27 | **11.15** |
| $\#C = 1$ | 1 | 10.78 | 11.90 | 11.07 | 11.34 | 11.83 | **12.95** |
| | 2 | 8.54 | 8.91 | 8.79 | 8.78 | 9.20 | **17.83** |
| | 0 | 15.19 | 20.14 | 16.11 | 15.99 | 22.95 | **31.89** |
| $\#C = 3$ | 1 | 22.27 | 22.64 | 23.42 | 24.38 | 32.74 | **33.02** |
| | 2 | 22.90 | 22.85 | 23.43 | 23.41 | 28.48 | **36.63** |

Table 8: Test accuracy of the server model of different methods in CIFAR10 in three random seeds and across five levels of statistical heterogeneity (lower $\alpha$ is more heterogeneous).

| $p \sim Dir(\alpha)$ | seed | FedAvg | FedFTG | DENSE | Co-Boosting | DeepInversion | FedMITR |
|---|---|---|---|---|---|---|---|
| | 0 | 10.32 | 11.02 | 10.33 | 10.37 | 12.77 | **20.49** |
| 0.01 | 1 | 13.83 | 16.09 | 14.66 | 14.59 | 15.49 | **20.57** |
| | 2 | 10.94 | 11.09 | 11.15 | 11.25 | 11.99 | **16.50** |
| | 0 | 13.85 | 14.75 | 14.69 | 14.74 | 19.59 | **30.60** |
| 0.05 | 1 | 18.99 | 18.98 | 19.10 | 20.03 | 24.47 | **31.68** |
| | 2 | 14.77 | 14.76 | 15.45 | 17.16 | 21.96 | **26.54** |
| | 0 | 27.63 | 28.59 | 28.19 | 29.71 | 36.01 | **40.37** |
| 0.1 | 1 | 24.98 | 26.28 | 25.41 | 26.68 | 33.25 | **35.69** |
| | 2 | 20.29 | 20.29 | 21.97 | 24.25 | 32.06 | **34.84** |
| | 0 | 41.87 | 45.43 | 42.64 | 45.72 | 46.21 | **52.56** |
| 0.3 | 1 | 37.47 | 39.88 | 39.77 | 41.34 | 47.28 | **54.01** |
| | 2 | 34.46 | 34.73 | 34.64 | 34.66 | 40.40 | **48.50** |
| | 0 | 37.81 | 41.52 | 38.13 | 39.04 | 42.63 | **43.37** |
| 0.5 | 1 | 39.42 | 39.49 | 40.46 | 42.82 | 46.69 | **49.91** |
| | 2 | 40.87 | 45.05 | 41.64 | 42.78 | 45.68 | **55.06** |

Table 9: Test accuracy of the server model of different methods in OfficeHome in three random seeds and across five levels of statistical heterogeneity (lower $\alpha$ is more heterogeneous).

| $p \sim Dir(\alpha)$ | seed | FedAvg | FedFTG | DENSE | Co-Boosting | DeepInversion | FedMITR |
|---|---|---|---|---|---|---|---|
| | 0 | 9.16 | 9.51 | 9.41 | 9.69 | 12.72 | **25.19** |
| 0.01 | 1 | 6.48 | 7.08 | 7.01 | 7.26 | 10.13 | **22.82** |
| | 2 | 8.54 | 9.79 | 9.45 | 9.51 | 12.78 | **24.13** |
| | 0 | 14.84 | 15.27 | 15.02 | 15.37 | 19.11 | **30.49** |
| 0.05 | 1 | 11.78 | 12.94 | 12.69 | 13.12 | 17.64 | **28.40** |
| | 2 | 14.43 | 15.62 | 15.27 | 15.62 | 19.42 | **31.23** |
| | 0 | 20.39 | 21.26 | 21.01 | 21.42 | 25.44 | **34.60** |
| 0.1 | 1 | 16.08 | 17.49 | 17.08 | 17.49 | 22.69 | **32.01** |
| | 2 | 16.43 | 18.55 | 17.39 | 17.55 | 23.53 | **31.83** |
| | 0 | 27.90 | 29.21 | 28.74 | 29.36 | 34.98 | **35.82** |
| 0.3 | 1 | 27.65 | 29.80 | 29.40 | 29.11 | 33.95 | **35.19** |
| | 2 | 25.09 | 27.37 | 26.59 | 27.34 | 31.70 | **35.57** |
| | 0 | 34.07 | 36.07 | 35.50 | 35.88 | **41.33** | 40.02 |
| 0.5 | 1 | 30.33 | 32.01 | 31.89 | 32.29 | **37.41** | 37.00 |
| | 2 | 29.02 | 30.49 | 30.05 | 30.55 | 34.88 | **37.44** |

Table 10: Test accuracy of the server model of different methods in Mini-ImageNet in three random seeds and across five levels of statistical heterogeneity (lower $\alpha$ is more heterogeneous).

| $p \sim Dir(\alpha)$ | seed | FedAvg | FedFTG | DENSE | Co-Boosting | DeepInversion | FedMITR |
|---|---|---|---|---|---|---|---|
| 0.01 | 0 | 12.01 | 12.47 | 12.53 | 13.14 | 20.86 | **40.83** |
| | 1 | 14.33 | 15.35 | 14.76 | 15.17 | 22.05 | **50.89** |
| | 2 | 15.62 | 16.36 | 16.18 | 16.93 | 24.56 | **44.07** |
| 0.05 | 0 | 40.41 | 41.20 | 40.89 | 41.74 | 50.49 | **62.26** |
| | 1 | 36.26 | 37.49 | 37.28 | 37.19 | 44.82 | **62.36** |
| | 2 | 37.28 | 38.07 | 37.79 | 38.17 | 46.72 | **61.83** |
| 0.1 | 0 | 54.87 | 55.69 | 55.33 | 55.83 | 62.71 | **69.94** |
| | 1 | 50.92 | 52.53 | 51.91 | 52.22 | 59.19 | **68.67** |
| | 2 | 51.64 | 52.65 | 52.32 | 52.36 | 58.95 | **66.01** |
| 0.3 | 0 | 75.78 | 76.69 | 76.19 | 76.49 | **78.84** | 76.58 |
| | 1 | 78.81 | 79.32 | 79.28 | 79.37 | **80.47** | 79.17 |
| | 2 | 75.92 | 76.90 | 76.48 | 76.51 | **77.88** | 76.56 |
| 0.5 | 0 | 81.83 | 82.09 | 82.08 | 82.09 | **82.42** | 81.93 |
| | 1 | 82.12 | 82.45 | 82.24 | 82.43 | **83.93** | 82.33 |
| | 2 | 81.67 | 82.22 | 82.17 | 82.10 | **82.93** | 82.27 |

Table 11: Test accuracy of the server model of different methods in Mini-ImageNet in three random seeds under extreme heterogeneity setting.

| $\#C = k$ | seed | FedAvg | FedFTG | DENSE | Co-Boosting | DeepInversion | FedMITR |
|---|---|---|---|---|---|---|---|
| $\#C = 10$ | 0 | 7.37 | 7.80 | 7.77 | 7.82 | 12.89 | **20.72** |
| | 1 | 8.24 | 9.54 | 9.02 | 9.24 | 14.72 | **23.13** |
| | 2 | 7.78 | 8.30 | 8.05 | 8.12 | 13.15 | **21.57** |

