# OpenReview forum: "Exploring One-Shot Federated Learning by Model Inversion and Token Relabel with Vision Transformers"
_ICLR.cc/2025/Conference — Submitted to ICLR 2025_

### Official Review · Reviewer_J6S5 · 2024-10-31

**Soundness:** 3
**Presentation:** 2
**Contribution:** 3
**Rating:** 3
**Confidence:** 4

**Summary:**

This paper investigates the problem of one-shot federated learning, which learn a global model by aggregating parameters learned from different local clients in a single communication. The global model is essentially a weighted ensemble of outputs from different models. The ensemble weights are learned by training from a set of synthetic data. However, the data quality is not satisfactory due to naïve generator and mismatch between data and label due to huge disparity between synthetic data and separate local training data on the clients. To address this problem, the method first uses the model inversion to synthesize training data from random noise via the pre-trained local models. Then, it uses high-quality patches of the synthesized images, i.e., patch with high information density, or low-density but relabeled, to train the parameters regarding ensemble in the global model.

**Strengths:**

1.	The problem setup is of high importance.
2.	The model inversion to generate the training data and distinguish training data at patch-level with different information density is interesting.
3.	The experimental study is conducted on many different datasets and the performance gain is significant.

**Weaknesses:**

1.	The paper writing is not clear and the method is hard to follow. In particular, how do you proceed to your method design from your analysis. In other words, what is the motivation of your approach based on your observation.
2.	 The performance is too good. Such a clear performance gain makes me worry about the fairness in comparison. More comparison details should be provided for clarification, including model size, training time, and training data. For example, the backbone model is pre-trained on ImageNet-1K but tested on dataset which is just a subset of ImageNet, e.g., Mini-ImageNet? I am not sure whether this is a common setup.
3.	How do you define and calculate the information density, a criteria used to determine the re-labeling.
4.	How many models (local clients) are trained in your experimental setup. I am curious whether weights are directly calculated from the input data (learnable parameters or predicted from the data).

**Questions:**

Please address all of my (4) questions in the weakness, in particular the first two.

---

### Official Review · Reviewer_ycQS · 2024-10-31

**Soundness:** 2
**Presentation:** 3
**Contribution:** 2
**Rating:** 6
**Confidence:** 3

**Summary:**

The paper proposes **FedMITR**, a novel One-Shot Federated Learning (FL) framework that employs model inversion and token relabeling to improve performance in highly heterogeneous data environments. The approach utilizes Vision Transformers (ViTs) and model inversion techniques to synthesize image data without access to original client data, protecting privacy. The synthesized data is further refined by re-labeling image patches based on their information density.

**Strengths:**

- The method for understanding and tackling the challenges of existing studies on One-shot FL is novel.
- The method does not rely on accessing client data, making it privacy-friendly. By generating synthetic data from random noise, it avoids direct exposure of sensitive information.
- A detailed experiment was conducted by reflecting recently researched models such as Co-Boosting [1].


[1] Dai, R., Zhang, Y., Li, A., Liu, T., Yang, X., & Han, B. (2024). Enhancing One-Shot Federated Learning Through Data and Ensemble Co-Boosting. arXiv preprint arXiv:2402.15070.

**Weaknesses:**

- It is thought to be a rather peripheral study, but the approach is novel.
- It is expected that the amount of computation of FedMITR is higher than other methods. Have you compared this?
- The results of the IID case need to be shared. It is necessary to share the results of the experiments using higher values of Dirichlet distribution parameters. $\alpha=0.5$ is still the data is heterogeneous to express IID.
- It is necessary to use a larger dataset for experiments. However, this is a chronic problem in federated learning studies, and it is not just a weakness of FedMITR.

**Questions:**

- There seems to be no need to share the results for each seed in the Appendix.
- Does FedMITR only work using ViTs?

---

> ### Comment · Reviewer_ycQS · 2024-12-02
>
> Q4. I hope authors compare it through the experimental results.
>
> Q5. I agree that the IID setting unrealistic, but it is not entirely worth experimenting with. IID experiments in federated learning are essential for establishing a baseline, validating theoretical assumptions, and understanding the algorithm's maximum potential performance before addressing real-world.
>
> Some of my concerns have been addressed, but not completely. I'll keep the score.

---

### Official Review · Reviewer_XJRk · 2024-11-04

**Soundness:** 2
**Presentation:** 3
**Contribution:** 2
**Rating:** 5
**Confidence:** 4

**Summary:**

This paper presents a federated model inversion and token relabel framework for one-shot federated learning. The generated pseudo labels are used to train the global model using patches with high information density.

**Strengths:**

+ Model inversion is first explored in one shot FL.
+ The proposed method achieves good results as compared several existing methods in the one shot FL setting.

**Weaknesses:**

- In Fig. 1, it is not clear what model inversion method is used and what is the model for inversion.
- The authors state the goal of model inversion is also not to leak any private information, however, there is no justification for this of the proposed method.
- The concept of “information density” of tokens is not clear, and why it can used for token selection. How will it affect the image reconstruction/synthesized quality?  This further makes the token relabeling process unclear.
- The datasets used in the experimental evaluation are relatively small. Large datasets such as ImageNet should be considered.
- Regarding the privacy aspect, although the authors provide visualization of the generated patches, it does not guarantee data privacy (privacy attributes can still be leaked from the reconstructed data/feature). Formal evaluation (e.g. performance under DP) is useful.
- More recently one-shot FL methods should be compared.
[A] Data-free one-shot federated learning under very high statistical heterogeneity
[B] Navigating Heterogeneity and Privacy in One-Shot Federated Learning with Diffusion Models

**Questions:**

Please refer to the weakness section.

---

### Meta-Review · Area_Chair_DuLN · 2024-12-24

**Metareview:**

The paper proposes FedMITR, a framework for one-shot federated learning using model inversion and token relabeling, designed to address challenges of heterogeneous data distributions in federated settings. By employing Vision Transformers (ViTs) and selectively relabeling patches with high information density, the method aims to synthesize higher-quality data and improve global model performance.

Reviewers appreciated the novelty of addressing one-shot federated learning with a focus on model inversion and token relabeling, recognizing the significance of the problem. However, significant concerns were raised about the method’s clarity, evaluation, and generalizability. Reviewers questioned the lack of clear motivation and justification for design choices, the limited dataset scope, and the insufficient exploration of privacy implications. Concerns about computational complexity and the applicability of the proposed information density metric were also highlighted. Despite the detailed rebuttal provided by the authors, reviewers maintained that critical concerns about clarity, evaluation completeness, and privacy remained unresolved.

The authors responded to the reviewers' questions, providing additional explanations about methodology and addressing some concerns, such as the definition of information density and comparison setups. However, key issues about experimental rigor, fairness in comparisons, and clarity of contributions persisted, as reflected in reviewers’ unchanged ratings.

Considering these factors, the AC recommends rejection. While the paper proposes an innovative approach to a challenging problem, unresolved concerns about experimental design, clarity, and broader applicability prevent it from being a strong contribution at this stage.

**Additional Comments On Reviewer Discussion:**

Please refer to the meta review.

---

### Decision · Program_Chairs · 2025-01-22

Reject